# Discovering General Reinforcement Learning Algorithms with Adversarial Environment Design

**Matthew T. Jackson**[*]
University of Oxford

**Minqi Jiang**
UCL

**Jack Parker-Holder**
Google DeepMind

**Risto Vuorio**
University of Oxford

**Chris Lu**
University of Oxford

**Gregory Farquhar**
Google DeepMind

**Shimon Whiteson**
University of Oxford

**Jakob N. Foerster**
University of Oxford

## Abstract

The past decade has seen vast progress in deep reinforcement learning (RL) on the back of algorithms manually designed by human researchers. Recently, it has been shown that it is possible to meta-learn update rules, with the hope of discovering algorithms that can perform well on a wide range of RL tasks. Despite impressive initial results from algorithms such as Learned Policy Gradient (LPG), there remains a generalization gap when these algorithms are applied to unseen environments. In this work, we examine how characteristics of the meta-training distribution impact the generalization performance of these algorithms. Motivated by this analysis and building on ideas from *Unsupervised Environment Design* (UED), we propose a novel approach for automatically generating curricula to maximize the *regret* of a meta-learned optimizer, in addition to a novel approximation of regret, which we name *algorithmic regret* (AR). The result is our method, General RL Optimizers Obtained Via Environment Design (GROOVE). In a series of experiments, we show that GROOVE achieves superior generalization to LPG, and evaluate AR against baseline metrics from UED, identifying it as a critical component of environment design in this setting. We believe this approach is a step towards the discovery of truly general RL algorithms, capable of solving a wide range of real-world environments.

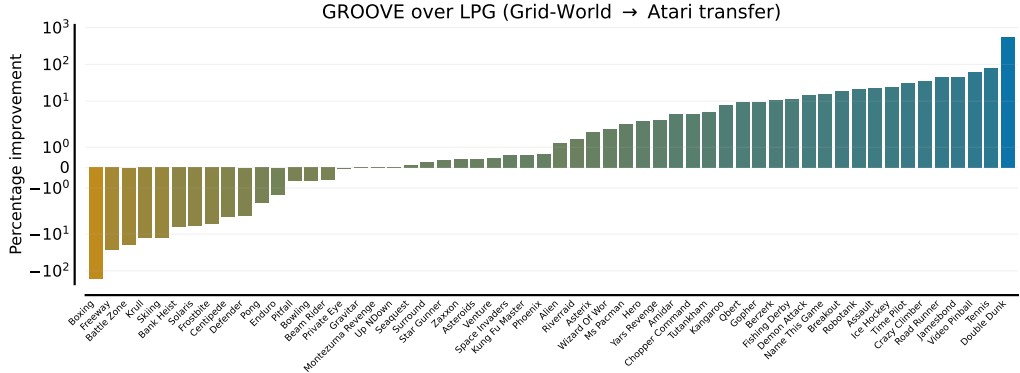

Figure 1: Out-of-distribution performance on Atari—after meta-training exclusively on Grid-World levels, our method (GROOVE) significantly outperforms LPG on Atari. Improvement is measured as a percentage of mean human-normalized return over 5 seeds.

[*]Correspondence to `jackson@robots.ox.ac.uk`.

37th Conference on Neural Information Processing Systems (NeurIPS 2023).

# 1 Introduction

The past decade has seen vast progress in deep reinforcement learning [Sutton and Barto, 1998, RL], a paradigm whereby agents interact with an environment to maximize a scalar reward. In particular, deep RL agents have learned to master complex games [Silver et al., 2016, 2017, Berner et al., 2019], control physical robots [OpenAI et al., 2019, Andrychowicz et al., 2020, Miki et al., 2022] and increasingly solve real-world tasks [Degrave et al., 2022]. However, these successes have been driven by the development of manually-designed algorithms, which have been refined over many years to tackle new challenges in RL. As a result, these methods do not always exhibit the same performance when transferred to new tasks [Henderson et al., 2018, Andrychowicz et al., 2021] and are limited by our intuitions for RL.

Recently, *meta-learning* has emerged as a promising approach for discovering general RL algorithms in a data-driven manner [Beck et al., 2023b]. In particular, Oh et al. [2020] introduced Learned Policy Gradient (LPG), showing it is possible to meta-learn an update rule on toy environments and transfer it zero-shot to train policies on challenging, unseen domains. Despite impressive initial results, there remains a significant generalization gap when these algorithms are applied to unseen environments. In this work, we seek to learn general and robust RL algorithms, by examining how characteristics of the *meta-training distribution* impact the generalization of these algorithms.

Motivated by this analysis, our goal is to automatically learn a meta-training distribution. We build on ideas from *Unsupervised Environment Design* [Dennis et al., 2020, UED], a paradigm where a student agent trains on an adaptive distribution of environments proposed by a teacher, which seeks to propose tasks which maximize the student's *regret*. UED has typically been applied to train single RL agents, where it has been shown to produce robust policies capable of zero-shot transfer to challenging human-designed tasks. Instead, we apply UED to the meta-RL setting of meta-learning a policy optimizer, which we refer to as *policy meta-optimization* (PMO). For this, we propose *algorithmic regret* (AR), a novel metric for selecting meta-training tasks, in addition to a method building on LPG and ideas from UED. We name our method *General **R**L **O**ptimizers **O**btained **V**ia **E**nvironment Design*, or GROOVE.

We train GROOVE on an unstructured distribution of Grid-World environments, and rigorously examine its performance on a variety of unseen tasks—ranging from challenging Grid-Worlds to Atari games. When evaluated against LPG, GROOVE achieves significantly improved generalization performance on all of these domains. Furthermore, we compare AR against prior environment design metrics proposed in UED literature, identifying it as a critical component for environment design in this setting. We believe this approach is a step towards the discovery of truly-general RL algorithms, capable of solving a wide range of real-world environments.

We implement GROOVE and LPG in JAX [Bradbury et al., 2018], resulting in a meta-training time of 3 hours on a single V100 GPU. As well as being the first complete and open-source implementation of LPG, we achieve a major speedup against the reference implementation, which required 24 hours on a 16-core TPU-v2. This will enable academic labs to perform follow-up research in this field, where compute constraints have long been a limiting factor.

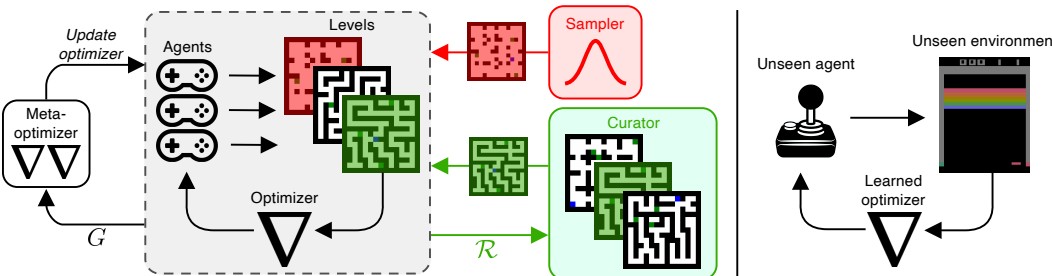

Figure 2: GROOVE meta-training (left) and meta-testing (right). During meta-training, levels are sampled from both the level curator and sampler. Agents are trained by an optimizer for multiple updates, before a meta-optimizer updates the optimizer based on agent return. At the end of an agent's lifetime, its regret is calculated and the level curator is updated. During meta-testing, the trained optimizer is applied to previously unseen environments and agent architectures.

Our contributions are summarized as follows:

- In order to distinguish this problem setting from traditional meta-RL, we provide a novel formulation of PMO using the Meta-UPOMDP (Section 2.1).
- We propose AR (Section 3.2), a novel regret approximation for PMO, and GROOVE (Section 3.3), a PMO method using AR for environment design.
- We analyze how features of the meta-training distribution impact generalization in PMO (Section 4.2) and demonstrate AR as a proxy for task informativeness (Section 4.3).
- We extensively evaluate GROOVE against LPG, demonstrating improved in-distribution robustness and out-of-distribution generalization (Section 4.4).
- We perform an ablation of AR, demonstrating the insufficiency of existing methods (PLR and LPG) without AR, as well as the impact of the antagonist agent in AR (Section 4.5).

## 2 Problem Setting and Preliminaries

### 2.1 Formulating Policy Meta-Optimization for Environment Design

**Policy Meta-Optimization**    In this work, we consider a subproblem of meta-RL which we refer to as *policy meta-optimization* (PMO). PMO is a bilevel optimization problem. In the *inner loop*, a collection of agents each interact with their associated environments and are updated with a policy optimizer. Following a series of inner-loop updates, the *outer loop* updates the policy optimizer in order to maximize the performance of these agents. PMO only trains the policy optimizer in the outer loop, while the initial agent parameters for each task are generated by a static initialization function.

**Unsupervised Environment Design**    In UED [Dennis et al., 2020], a *teacher* is given the problem of designing an environment distribution which is maximally useful for training a *student* agent. UED formalizes this problem setting with the Underspecified Partially Observable Markov Decision Process (UPOMDP), an extension of the POMDP with additional *free parameters* $\phi$ that parameterize those aspects of the environment which the teacher can modify throughout training. In prior work, this paradigm has been used to adapt environment distributions to facilitate the learning of a robustly transferable policy [Jiang et al., 2021b, Parker-Holder et al., 2022a]. However, in our problem setting of PMO, the central focus is on learning the update rule itself, with the goal of transferring the update rule to new environments.

**Problem Formulation**    We therefore introduce the *Meta-UPOMDP*, an extension of the UPOMDP, to account for this difference. Formally, the Meta-UPOMDP is defined by the tuple $\langle \mathcal{A}, \mathcal{O}, \mathcal{S}, \mathcal{T}, \mathcal{I}, \mathcal{R}, \gamma, \Phi, \Theta \rangle$. The first components correspond to the standard UPOMDP, where $\mathcal{A}$ is the action space, $\mathcal{S}$ is the state space, $\mathcal{O}$ is the observation space, and $\mathcal{T} : \mathcal{S} \times \mathcal{A} \times \Phi \mapsto \mathcal{S}$ is the transition function. Upon each transition, the student agent receives an observation according to the observation function $\mathcal{I} : \mathcal{S} \mapsto \mathcal{O}$ and a reward according to the reward function $\mathcal{R} : \mathcal{S} \times \mathcal{A} \mapsto \mathbb{R}$. Here, the free parameters $\phi \in \Phi$ control the variable aspects of the environment, such as the $x, y$-positions of obstacles in a 2D maze.[1]

The Meta-UPOMDP models PMO, in which an optimizer $\mathcal{F} : \Theta \times \mathbb{T} \mapsto \Theta$ learns to update an agent's parameters $\Theta$ given the sequence of states, actions, rewards and termination flags corresponding to the agent's past experience $\mathbb{T}$ in the environment. This update is performed after every transition over the agent's *lifetime* of $N$ environment interactions, resulting in a sequence of parameters $(\theta_0, \ldots, \theta_N)$. The Meta-UPOMDP extends the UPOMDP include agent parameters $\Theta$ and appends the agent lifetime $N$ to the free parameters $\phi$, making it a controllable feature of tasks.

For an initialization of agent parameters $\theta_0$ and free parameters $\phi$, we define the value of the optimizer to be $V_{\phi,\theta_0}(\mathcal{F}) = \mathbb{E}_{\pi_{\theta_N}}[\sum_t^\infty \gamma^t r_t]$, which is the expected return of the *trained agent* $\pi_{\theta_N}$ on the environment specified by $\phi$ at the end of its lifetime. Given an optimizer $\mathcal{F}_\eta$ with meta-parameters $\eta$, we reformulate the PMO objective from Oh et al. [2020] to

$$\mathcal{L}(\eta) = \mathbb{E}_{\phi \sim p(\phi)} \mathbb{E}_{\theta_0 \sim p(\theta_0)} [V_{\phi,\theta_0}(\mathcal{F}_\eta)], \tag{1}$$

where $p(\phi)$ and $p(\theta_0)$ are distributions of free parameters and initial agent parameters.

---

[1] The terms "tasks" and "levels" are used interchangeably to refer to settings of free parameters.

## 2.2 Learned Policy Gradient

Learned Policy Gradient [Oh et al., 2020, LPG] is a PMO method which trains a generalization of the actor-critic architecture [Barto et al., 1983]. This replaces the critic with a generalization of value functions from RL, that we refer to as *bootstrap functions*. Whilst value functions are trained to predict the expected discounted return from a given state, bootstrap functions predict an $n$-dimensional, categorical *bootstrap vector*, the properties of which are meta-learned by LPG.

LPG uses a reverse-LSTM [Hochreiter and Schmidhuber, 1997] to learn a policy update for each agent transition, conditioned on all future episode transitions. For a single update to agent parameters $\theta$ at time-step $t$, LPG outputs targets $\hat{y}_t, \hat{\pi}_t = U_\eta(x_t|x_{t+1}, \ldots, x_T)$, where $x_t = [r_t, d_t, \gamma, \pi_\theta(a_t|s_t), y_\theta(s_t), y_\theta(s_{t+1})]$ is a vector containing reward $r_t$, episode-termination flag $d_t$, discount factor $\gamma$, probability of the chosen action $\pi_\theta(a_t|s_t)$, and bootstrap vectors for the current and next states $y_\theta(s_t)$ and $y_\theta(s_{t+1})$. The targets $\hat{y}$ and $\hat{\pi}$ update the bootstrap function and policy respectively, giving the update rule

$$\Delta\theta \propto [\nabla_\theta \log \pi_\theta(a|s)\hat{\pi} - \alpha_y \nabla_\theta D_{\text{KL}}(y_\theta||\hat{y})]. \tag{2}$$

## 2.3 Learning Robust Policies via Minimax-Regret UED

A trivial example of UED is domain randomization [Jakobi, 1997, DR], in which the teacher generates an environment distribution by uniformly sampling free parameters from the UPOMDP. By sampling randomly, DR often fails to generate environments with interesting structure: they may be trivial, impossible to solve, or irrelevant to downstream tasks of interest. An alternative approach is to train an adversarial *minimax* teacher, whose objective is to generate environments which minimize the agent's return. This has the benefit of adapting the environment distribution to the agent's current capability, by presenting it with environments on which it performs poorly. However, by naïvely minimizing return, the teacher is incentivized to generate environments which are impossible for the agent to solve.

In contrast to this, Dennis et al. [2020] propose the *minimax-regret* objective for UED, in which the teacher aims to maximize the agent's regret, the difference between the achieved and maximum return. Unlike return minimization, this disincentivizes the teacher from generating unsolvable levels where regret would be $0$. However, since computing the maximum return is generally intractable, methods implementing this objective have proposed approximations of regret. PAIRED [Dennis et al., 2020] co-trains an *antagonist* agent with the original (*protagonist*) agent, estimating regret as the difference in their performance. PLR [Jiang et al., 2021b,a] curates a buffer of high-regret levels, rather than training a generative model to produce them. In this, a range of regret approximations are evaluated, with positive value loss, L1 value loss, and maximum Monte Carlo achieving consistent performance across the evaluated domains.

# 3 Adversarial Environment Design for Policy Meta-Optimization

In this section, we introduce GROOVE, a novel method for PMO. We begin by formulating the minimax-regret objective implemented by GROOVE, followed by AR, a novel regret approximation designed for PMO. Finally, we discuss existing approaches to environment design and motivate the selection of a curation-based approach.

## 3.1 Minimax-Regret as a Meta-Objective

Our method trains an optimizer $\mathcal{F}_\eta$ over an adversarial distribution of environments, in which the UED adversary's objective is to maximize the *regret* of the meta-learner, defined as

$$\text{REGRET}(\eta, \phi) = V_\phi(\eta_\phi^*) - V_\phi(\eta). \tag{3}$$

Here, $\eta_\phi^*$ denotes the optimal meta-parameters for updating the student's policy on an environment instance $\phi$, i.e., $\eta_\phi^* = \arg\max_\eta \mathbb{E}_\phi \mathbb{E}_{\theta_0 \sim p(\theta_0)}[V_{\phi,\theta_0}(\mathcal{F}_\eta)]$. The function $V_\phi : \mathcal{H} \mapsto \mathbb{R}$ defines the expected return of the student policy on $\phi$ at the end of the its lifetime (after $N$ updates) using the update rule parameterized by $\eta \in \mathcal{H}$, when trained from an initialization $\theta_0 \sim p(\theta_0)$.

## 3.2 Approximating Minimax-Regret

As in the traditional UED setting, regret is generally intractable to compute. A range of scoring functions have been proposed to approximate regret, often deriving the approximation from value loss [Jiang et al., 2021b,a]. Two consistently high performing metrics are L1 value loss and positive value loss, which are equal to the episodic mean of absolute and positive GAE [Schulman et al., 2016] terms respectively.

In order to exploit the structure of our problem setting, we propose an alternative approximation which we refer to as *algorithmic regret* (AR). In this, we co-train an *antagonist* agent using a manually designed RL algorithm $\mathcal{A}$ (e.g., A2C [Mnih et al., 2016], PPO [Schulman et al., 2017]) in parallel to the *protagonist* agent trained by GROOVE. AR is then computed from the difference in final performance against the antagonist,

$$\text{REGRET}^{\mathcal{A}}(\eta, \phi) = V_\phi(\mathcal{A}) - V_\phi(\eta) \tag{4}$$

where $V_\phi(\mathcal{A})$ denotes the expected return of the antagonist policy when trained with $\mathcal{A}$.

We evaluate AR against both L1 value loss and positive value loss in Section 4.4, demonstrating improved generalization performance on Min-Atar.

## 3.3 Environment Design

Following the dual-curriculum design paradigm from Jiang et al. [2021a], a large class of UED methods can be represented as a combination of two teachers: a *curator* and a *generator*. Here, the level generator is a generative model that is optimized to produce regret-maximizing levels, whilst the level curator maintains a set of previously-visited high-regret levels to be replayed by the agent. The generator provides a slowly adapting mechanism for environment design, allowing the method to design new levels without random sampling, whilst the curator provides a quickly-adapting replay buffer of useful levels.

In PMO, a single sample of environment regret requires an agent to be trained to convergence and subsequently evaluated on that environment. This is significantly less sample efficient than traditional UED, where regret is measured from a single rollout of the current policy. Furthermore, generator-based methods for UED (e.g. PAIRED) have been shown to achieve lower performance and sample efficiency than curation-based approaches [Jiang et al., 2021a, Parker-Holder et al., 2022a]. Due to this, we design GROOVE using PLR (Section 2.3), which curates randomly generated levels, avoiding the need to train a level generator.

The meta-training loop for GROOVE is presented in Algorithm 1 and Figure 2.

---

**Algorithm 1** GROOVE meta-training

---

    **input**: Environment set $\Phi$, agent parameter initialization function $p(\theta)$
    **initialize**: Meta-parameters $\eta$, PLR level buffer $\Lambda$, agent-environment lifetimes $\{\theta, \phi\}_i$
    **repeat**
       **for all** lifetimes $\{\theta, \phi\}_i$ **do**
          **for** update $\leftarrow 1$ to $K$ **do**
             Rollout agent $\pi_\theta$ on environment $\phi$
             Update $\theta$ with $\mathcal{F}_\eta$ (Equation 2)
          **end for**
          Compute meta-gradient for $\eta$ with updated $\theta$ (Equation 1)
          **if** lifetime over **then**
             Evaluate regret approximation $\mathcal{R} \leftarrow \text{REGRET}^*(\eta, \phi)$ with final $\theta$
             Update level buffer $\Lambda$ with $\mathcal{R}$ and $\phi$
             Reinitialize lifetime $(\theta, \phi) \sim p(\theta) \times \Lambda$
          **end if**
       **end for**
       Update $\eta$ with accumulated meta-gradients
    **until** $\eta$ converges

---

# 4    Experiments

Our experiments are designed to determine (1) how the meta-training distribution impacts OOD generalization in PMO, (2) how well AR identifies informative levels for generalization, and (3) the effectiveness of GROOVE at generating curricula for generalization using this metric. To achieve this, we first manually examine how informative and diverse meta-training distributions improve the generalization performance of LPG (Section 4.2). We then evaluate curricula generated by AR against those generated by randomly sampling and handcrafting (Section 4.3), demonstrating the ability to identify informative levels. Following this, we evaluate GROOVE against LPG (Section 4.4), demonstrating the impact of environment design for improving both in-distribution robustness and generalization performance. Finally, we evaluate GROOVE with AR against baseline metrics from the UED literature (Section 4.5), showing only AR consistently improves generalization.

## 4.1    Experimental Setup

**Training Environment**    For meta-training, we use a generalization of the tabular Grid-World environment presented by Oh et al. [2020]. In this environment distribution, a task is specified by the maximum episode length, grid size, wall placement, start position, and number of objects, whilst the objects themselves vary in position, reward, and probabilities of respawning or terminating the episode. This space contains tasks encapsulating thematic challenges in RL, including exploration, credit assignment, and stochasticity.

However, these challenges are notably sparse over the environment distribution. For instance, maze-like arrangements of walls induce a hard exploration challenge by creating anomalously long shortest-path lengths to objects, but are rare under uniform sampling. This captures the need for environment design, in order to discover complex and informative structures required for generalization.

**Testing Environments**    The purpose of our evaluation is to determine the generalization performance of the algorithms we consider, i.e., the expected return on real-world RL tasks. In order to approximate this, we evaluate on Atari [Bellemare et al., 2013], an archetypal RL benchmark, as well as its simplified counterpart Min-Atar [Young and Tian, 2019] for our intermediate results.

**Model Architecture and Implementation**    For our learned optimizer, we use the model architecture proposed in LPG. Since GROOVE is agnostic to the underlying meta-optimization method, we select LPG due to its state-of-the-art generalization performance on unseen tasks, in addition to the prior analysis of training distribution performed on LPG, which we build upon in Section 4.2. Further comparison to prior meta-optimization methods is presented in Section 5. Our experiments were executed on two to five servers, containing eight GPUs each (ranging in performance from 1080-Ti to V100). Model hyperparameters can be found in the supplementary materials and the project repository is available at https://github.com/EmptyJackson/groove.

## 4.2    Designing Meta-Training Distributions for Generalization

Unlike in standard RL, the impact of meta-training distributions on OOD generalization in PMO has not been explored in depth. One attempt at this came from Oh et al. [2020], who evaluate the generalization performance of LPG after meta-training on three different environment sets, varying both the number of tasks and the environments that the tasks are sampled from. By demonstrating improved transfer to Atari, the authors claim that two factors improve generalization performance:

1. Task *diversity* (the number of training tasks), and
2. How *informative* the tasks are for generalization.[2]

Whilst these results suggest a relationship between the meta-training distribution and generalization performance, the strength of these claims is limited by the number of environment sets (three) and confounding of these factors.

---

[2]Quote: "specific types of training environments... improved generalization performance." We refer to these types of environments as being *informative* for generalization.

To analyze the impact of task diversity, we sample Grid-World subsets of various sizes, before meta-training LPG on each of these and evaluating on Min-Atar (Figure 3). By generating environment sets with i.i.d. sampling from the Grid-World environment distribution, we remove the informativeness of tasks as a confounding factor. We observe a significant ($p < 0.05$) positive correlation in performance with number of levels in the aggregate task return, supporting the first claim. Furthermore, when considering the per-task breakdown of results (see supplementary materials), we observe a significant positive correlation on three of the four tasks, with the remaining task having a weak positive correlation, thereby supporting the first claim.

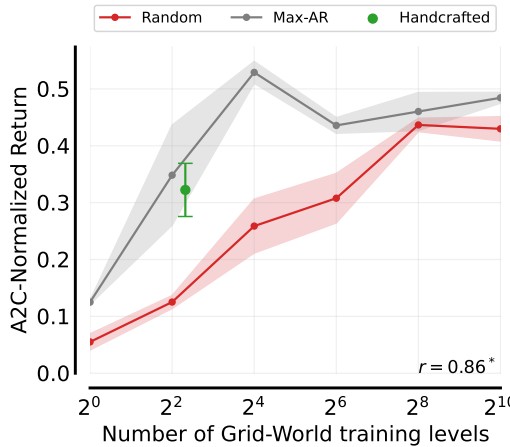

Figure 3: Aggregate performance on Min-Atar, with standard error shaded over 5 independent runs—PMCC is given for Random levels. A per-task breakdown is provided in the supplementary materials.

We investigate the second factor using a set of handcrafted Grid-World configurations proposed by Oh et al. [2020] (see supplementary materials). These are manually designed to emphasize stochasticity and credit assignment, two key challenges in RL, making them more informative for generalization than randomly sampled Grid-World configurations. At the same number of levels, we observe an improvement in performance from meta-training on handcrafted levels against random levels. Moreover, training on five handcrafted levels exceeds the performance of $2^6 = 64$ random levels, demonstrating the need for task distributions to contain tasks that are both informative and diverse.

## 4.3 Algorithmic Regret Identifies Informative Levels for Generalization

In Section 3.2, we hypothesize that AR identifies informative levels for meta-training. Before evaluating auto-curricula generated with this metric (Section 4.4), we evaluate static curricula generated by AR against random and handcrafted curricula. To achieve this, we train an LPG instance to convergence and collect a buffer of 10k unseen levels, ranked by the final AR of the model. We then train a new LPG instance on the highest-scoring levels, controlling for task diversity by subsampling variable-sized level sets from the buffer.

At all sizes of training environment set, we observe improved OOD transfer performance when training on high-AR levels compared to training on the same number of random levels (Figure 3). Furthermore, training with high-AR levels outperforms training over the same number of handcrafted levels, and far exceeds the performance of the fixed handcrafted set as the number of levels is increased, without requiring any human curation. These results validate AR as an effective metric for automatically generating informative curricula.

However, we note that the performance gap between high-AR and random levels decreases as the number of levels grows. This is not surprising, since the high-AR levels are generated by ranking and selecting the levels with the highest AR for each training set size, leading to a natural dilution in average AR. This also highlights the need for automatic curriculum generation throughout meta-training rather than training on static curricula, which we examine further in Section 4.4.

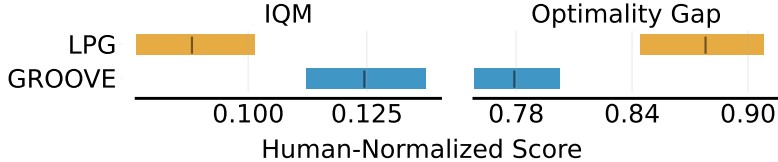

Figure 4: Aggregate performance metrics on Atari—shaded area shows 95% stratified bootstrap confidence interval (CI) over 5 seeds, following methodology from Agarwal et al. [2021]. Higher score is better for IQM and lower score is better for optimality gap.

## 4.4 Adversarial Environment Design Improves Generalization

We now evaluate the impact of environment design on generalization performance by comparing GROOVE to LPG, thereby evaluating the same meta-optimizer with and without environment design. After meta-training both methods on Grid-World, we first evaluate on randomly-sampled, unseen Grid-Worlds (Figure 5). In addition to GROOVE achieving higher mean performance, we observe increased robustness with GROOVE consistently outperforming LPG on their lower-scoring half of tasks. Notably, LPG fails to achieve greater than 75% A2C-normalized return on 187% more tasks than GROOVE. Despite this, GROOVE and LPG achieve comparable performance on their higher-scoring half of tasks, suggesting that GROOVE increases in-distribution robustness without reducing performance on easy tasks.

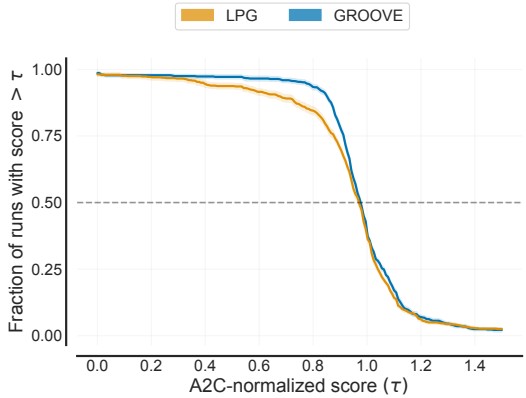

Figure 5: A2C-normalized score distribution on unseen Grid-World levels, shaded area shows a 95% CI over 10 seeds.

To evaluate generalization to challenging environments, we evaluate GROOVE against LPG on the Atari benchmark. GROOVE achieves superior per-task performance to LPG, achieving higher mean score on 39 vs. 17 tasks (Figure 1), with equal performance on one task (Montezuma's Revenge). Comparing aggregate performance, we observe significant increases in both IQM and optimality gap from GROOVE against LPG (Figure 4). While both of these methods achieve inferior performance to state-of-the-art, manually-designed RL algorithms [Hessel et al., 2018], the improvement from GROOVE highlights the importance of the meta-training distribution and potential for UED-based approaches when generalizing to complex and unseen environments.

## 4.5 Algorithmic Regret Outperforms Existing Metrics

Finally, we evaluate the quality of our proposed environment design metric, AR, against existing metrics from UED (Figure 6). We compare to L1 value loss and positive value loss (Section 3.2) due to their consistent performance in prior UED work, as well as regret against an optimal policy (as is analytically computable on Grid-World), which serves as an upper bound on regret. As a baseline, we also evaluate uniform scoring, which is equivalent to domain randomization (i.e. standard LPG).

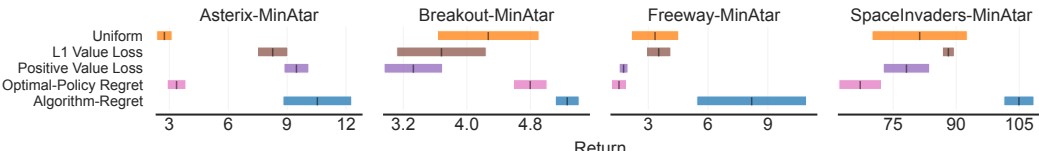

Figure 6: Evaluation of our proposed environment score function, AR, against uniform scoring, optimal-policy regret and baselines from UED literature. Mean return and standard error over 10 random seeds are shown.

AR achieves the highest performance on all tasks, significantly outperforming optimal-policy regret on all tasks and each of the other baselines on at least two out of four tasks. Furthermore, the value-loss metrics only significantly outperform uniform scoring on a single task, with positive value loss underperforming it on all others. This failure to identify informative levels for generalization highlights the challenge in transferring existing UED methods to PMO and the effectiveness of AR.

Whilst it is surprising that optimal-policy regret, which uses privileged level information, underperforms AR with an A2C antagonist, we hypothesize that the non-optimal performance and generality of handcrafted algorithms is a benefit for identifying informative levels. Optimal-policy regret does not account for training time, making it equivalent to an antagonist optimizer which always returns the optimal policy parameters, a setting likely to identify artificially difficult levels. To investigate this, we perform a comparison of A2C, PPO, random and expert antagonist agents (see supplementary materials), finding that using A2C or PPO antagonists outperforms random or expert agents.

# 5 Related Work

In this work, we examine the impact of training distribution for meta-learning general RL algorithms. More broadly, our work aligns with the "AI-generating algorithm" [Clune, 2019, AI-GA] paradigm, building upon two of the three pillars (learning algorithms and environments). In this section, we outline prior work in each of these fields and explain their relation to this work.

**Policy Meta-Optimization**    A common approach to PMO is to optimize RL algorithm components via meta-gradients [Xu et al., 2018, 2020]. In particular, ML[3] [Bechtle et al., 2021] uses meta-gradients to optimize domain-specific loss functions that transfer between similar continuous control tasks. MetaGenRL [Kirsch et al., 2020] expands upon this to optimize general loss functions that transfer to unseen tasks. LPG [Oh et al., 2020] discovers a general update rule that transfers from simple toy tasks to Atari environments. We choose to focus on LPG since it displays radical out-of-distribution transfer and imposes minimal structural bias on the learned update rule.

An alternative approach uses Evolution Strategies [Rechenberg, 1978, Salimans et al., 2017] to optimize RL objectives. EPG [Houthooft et al., 2018] evolves an objective function parameterized by a neural network that transfers to similar MuJoco environments, whilst DPO [Lu et al., 2022] evolves an objective that transfers from continuous control tasks to MinAtar environments. Other approaches to PMO *symbolically* evolve RL optimization components such as the loss function [Co-Reyes et al., 2021, Garau-Luis et al., 2022] or curiosity algorithms [Alet et al., 2020]. PMO is an instance of Auto-RL [Parker-Holder et al., 2022b], which automates the discovery of learning algorithms.

**Alternative Approaches to Meta-Reinforcement Learning**    PMO belongs to the subclass of *many-shot* meta-RL algorithms, which pose the setting of learning-to-learn given a substantial number of inner-loop environment interactions. An alternative class of approaches to this learn "intrinsic rewards", which augment the RL objective in order to improve learning. Alet et al. [2020] achieve this by meta-learning a program to transform the agent's objective, whilst Veeriah et al. [2021] propose a hierarchical method which meta-learns transferable options.

However, the majority of work in meta-RL has instead been on few-shot learning [Beck et al., 2023b]. RL[2] [Duan et al., 2016, Wang et al., 2016] use a black-box model for this, by representing both the policy and update rule with a recurrent neural network. A range of extensions to RL[2] have been proposed, which augment the original model with auxiliary task-inference objectives [Humplik et al., 2019, Zintgraf et al., 2020], additional exploration policies [Liu et al., 2021] and hypernetwork-based updates [Beck et al., 2023a]. Parameterized policy gradients are an alternative approach, with Model-Agnostic Meta-Learning [Finn et al., 2017, MAML] being the seminal method. MAML meta-learns a shared neural network initialization, such that it rapidly adapts to new tasks when optimized with policy gradients in the inner loop. Follow up work to this has proposed partitioning parameters [Zintgraf et al., 2019], modulating parameters for multimodal distributions [Vuorio et al., 2019] and investigated the reasons for its effectiveness [Raghu et al., 2019].

**Unsupervised Environment Design**    Unsupervised Environment Design (UED) was first proposed by Dennis et al. [2020] with the introduction of PAIRED, which trains a level generator for a single agent with minimax regret. GROOVE is based on PLR [Jiang et al., 2021b,a], which builds upon this objective by instead *curating* a buffer of high-regret levels. PLR remains one of the state of the art UED algorithms, which has been extended to consider multi-agent settings [Samvelyan et al., 2023], curriculum-induced covariate shift [Jiang et al., 2022] and more open-ended environment generators [Parker-Holder et al., 2022a]. Aside from regret, environments can also be selected to induce diversity in a population of agents [Brant and Stanley, 2017], most famously in the POET algorithm [Wang et al., 2019] which evolves a population of highly capable specialist agents.

UED falls more broadly into the field of open-endedness [Soros and Stanley, 2014], which attempts to design algorithms that continually produce novel and interesting behaviours. We take a step towards more open-ended algorithms by combining UED with meta-learning, thus discovering both algorithms and environments in a single method. Previous works combining these two AI-GA pillars include Team et al. [2023] who introduce a memory-based agent capable of human-timescale adaptation, and OpenAI et al. [2019] who train a policy capable of sim-to-real transfer to control a Rubik's Cube. Unlike our work, neither of these meta-learn general RL algorithms that can transfer to far out-of-distribution environments such as Atari.

# 6 Conclusion and Limitations

In this paper, we provide the first rigorous examination of the relationship between meta-training distribution and generalization performance in policy meta-optimization (PMO). Based on this analysis we leverage ideas from Unsupervised Environment Design (UED) and propose GROOVE, a novel method for PMO, in addition to a novel environment design metric, *algorithmic regret* (AR). Evaluating against LPG, we demonstrate significant gains in generalization from Grid-Worlds to Atari, as well as increased robustness to challenging in-distribution tasks. Finally, we identify AR as a critical component for applying environment design to PMO, demonstrating its effectiveness on Min-Atar, where prior UED metrics fail to outperform random sampling.

We acknowledge limitations in our work, largely driven by computational constraints. Given the huge number of environment interactions during meta-training, we heavily leverage our GPU-based Grid-World implementation in lowering training time. This limits our analysis to these fast but simple environments, meaning our conclusions cannot be guaranteed to generalize to more complex environments. Due to the cost of meta-testing on the large-scale Atari benchmark, our evaluation is also limited in variety of benchmarks, however we believe the diversity of the tasks in this domain sufficiently captures a range of challenges in RL.

By releasing our implementation—which is capable of meta-training these models on single GPU in hours, rather than days—we hope to spawn future work in this area from academic labs. In particular, these results may be scaled to training distributions with more complex and diverse environments, leading to the discovery of increasingly general RL algorithms.

## Acknowledgments and Disclosure of Funding

The authors thank Junhyuk Oh, Tim Rocktäschel and the anonymous NeurIPS reviewers for their helpful feedback that improved our paper. Matthew Jackson is funded by the EPSRC Centre for Doctoral Training in Autonomous Intelligent Machines and Systems, and Amazon Web Services.

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

# A   Summary of Notation

Table 1: Summary of notation used in this work.

| Symbol | Definition |
|---:|---|
| | **Meta-UPOMDP** |
| $\mathcal{A}, \mathcal{S}, \mathcal{O}$ | Action, state and observation spaces. |
| $\mathcal{T}, \mathcal{I}, \mathcal{R}$ | Transition, observation and reward functions. |
| $\gamma$ | Discount factor. |
| $\phi \in \Phi$ | Free parameters of the environment. |
| $\theta \in \Theta$ | Agent parameters, shared between actor and critic/bootstrap function. |
| $\mathbb{T}$ | Sequence of transitions $(\mathcal{O} \times \mathcal{A} \times \mathcal{R} \times \mathcal{O})$, denoted task "experience". |
| | **Policy Meta-Optimization** |
| $\mathcal{F}_\eta : \Theta \times \mathbb{T} \to \Theta$ | Agent optimizer. |
| $\eta \in \mathcal{H}$ | Agent optimizer parameters. |
| $V_{\phi,\theta_0}(\mathcal{F}_\eta)$ | Expected return of $\mathcal{F}_\eta$ at the end of training, given task $\phi$ and agent initialization $\theta_0$. |
| $V_\phi(\eta)$ | (Shorthand) Expected return of $\mathcal{F}_\eta$ at the end of training on task $\phi$, over a distribution of agent initializations $p(\theta_0)$. |
| | **Learned Policy Gradient** [Oh et al., 2020] |
| $\pi_\theta : \mathcal{A}, \mathcal{O} \to [0,1]$ | Agent policy. |
| $y_\theta : \mathcal{O} \to [0,1]^n$ | Agent bootstrap function—a generalization of value critics from RL, outputting a vector with semantics determined by the learned optimizer. |
| $U_\eta : \mathbb{T} \to [0,1]^n \times \mathbb{R}$ | LPG target function, outputting bootstrap function and policy targets $\hat{y}$ and $\hat{\pi}$ at time step $t$, conditioned on all future transitions. |

# B Hyperparameters

## B.1 GROOVE

Hyperparameters shared between GROOVE and LPG were tuned using LPG on Grid-World, then transferred to GROOVE without further tuning. The additional GROOVE hyperparameters (regarding the level buffer) were then tuned separately on Grid-World.

Table 2: GROOVE/LPG hyperparameters

| Hyperparameter | Value |
|---|---|
| Optimizer | Adam |
| Learning rate | 0.0001 |
| Discount factor | 0.99 |
| Policy entropy coefficient ($\beta_0$) | 0.05 |
| Bootstrap entropy coefficient ($\beta_1$) | 0.001 |
| L2 regularization coefficient for $\hat{\pi}$ ($\beta_2$) | 0.005 |
| L2 regularization coefficient for $\hat{y}$ ($\beta_3$) | 0.001 |
| Level buffer size | 4000 |
| Replay probability | 0.5 |
| Number of interactions per agent update | 20 |
| Number of agent updates per optimizer update | 5 |
| Number of parallel lifetimes | 512 |
| Number of parallel environments per lifetime | 64 |
| Algorithmic regret baseline algorithm | A2C |

## B.2 Agents

Agent hyperparameters were based on tuned A2C agents, before being fine-tuned with LPG. Since we meta-train on a continuous distribution of Grid-World environments, we do not use the agent hyperparameter bandit proposed by Oh et al. [2020] for meta-training.

Table 3: Agent hyperparameters—architecture descriptions $D(N)$ and $C(N)$ respectively refer to dense and convolutional layers of size $N$; ReLU activations are used throughout.

| Hyperparameter | Environment | | |
|---|---|---|---|
| | Grid-World | Min-Atar | Atari |
| Architecture | Tabular | D(64)-D(64) | C(32)-C(64)-C(64)-D(512) |
| Optimizer | SGD | Adam | Adam |
| Learning rate | 40 | 0.0005 | 0.0005 |
| Bootstrap KL coefficient ($\alpha_y$) | 0.5 | 0.5 | 0.5 |
| Train steps | 2500 | 100,000 | 100,000 |
| Agent seeds per LPG seed | 64 | 16 | 1 |

# C Handcrafted Environments

For our handcrafted environment set, we use the set of five tabular Grid-World configurations from Oh et al. [2020]. Grid-World objects are defined by $[r, \epsilon_{\text{term}}, \epsilon_{\text{respawn}}]$, where $r$ represents the reward when collected, $\epsilon_{\text{term}}$ is the episode-termination probability and $\epsilon_{\text{respawn}}$ is the probability of the object respawning each step after collection.

## C.1 Dense

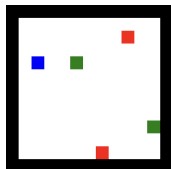

| Property | Value |
|---:|---|
| Size | $11 \times 11$ |
| Objects | $2 \times [1, 0, 0.05], [-1, 0.5, 0.1], [-1, 0, 0.5]$ |
| Maximum episode length | 500 |

## C.2 Sparse

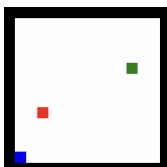

| Property | Value |
|---:|---|
| Size | $13 \times 13$ |
| Objects | $[1, 1, 0], [-1, 1, 0]$ |
| Maximum episode length | 50 |

## C.3 Long Horizon

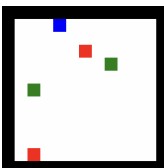

| Property | Value |
|---:|---|
| Size | $11 \times 11$ |
| Objects | $2 \times [1, 0, 0.01], 2 \times [-1, 0.5, 1]$ |
| Maximum episode length | 1000 |

## C.4 Longer Horizon

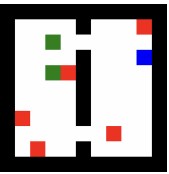

| Property | Value |
|---:|---|
| Size | $9 \times 9$ |
| Objects | $2 \times [1, 0.1, 0.01], 5 \times [-1, 0.8, 1]$ |
| Maximum episode length | 2000 |

Note: size is increased from $7 \times 9$ for consistency with our generalized Grid-World distribution.

## C.5 Long Dense

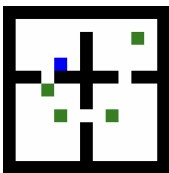

| Property | Value |
|---:|---|
| Size | $11 \times 11$ |
| Objects | $4 \times [1, 0, 0.005]$ |
| Maximum episode length | 2000 |

# D  Atari Training Curves

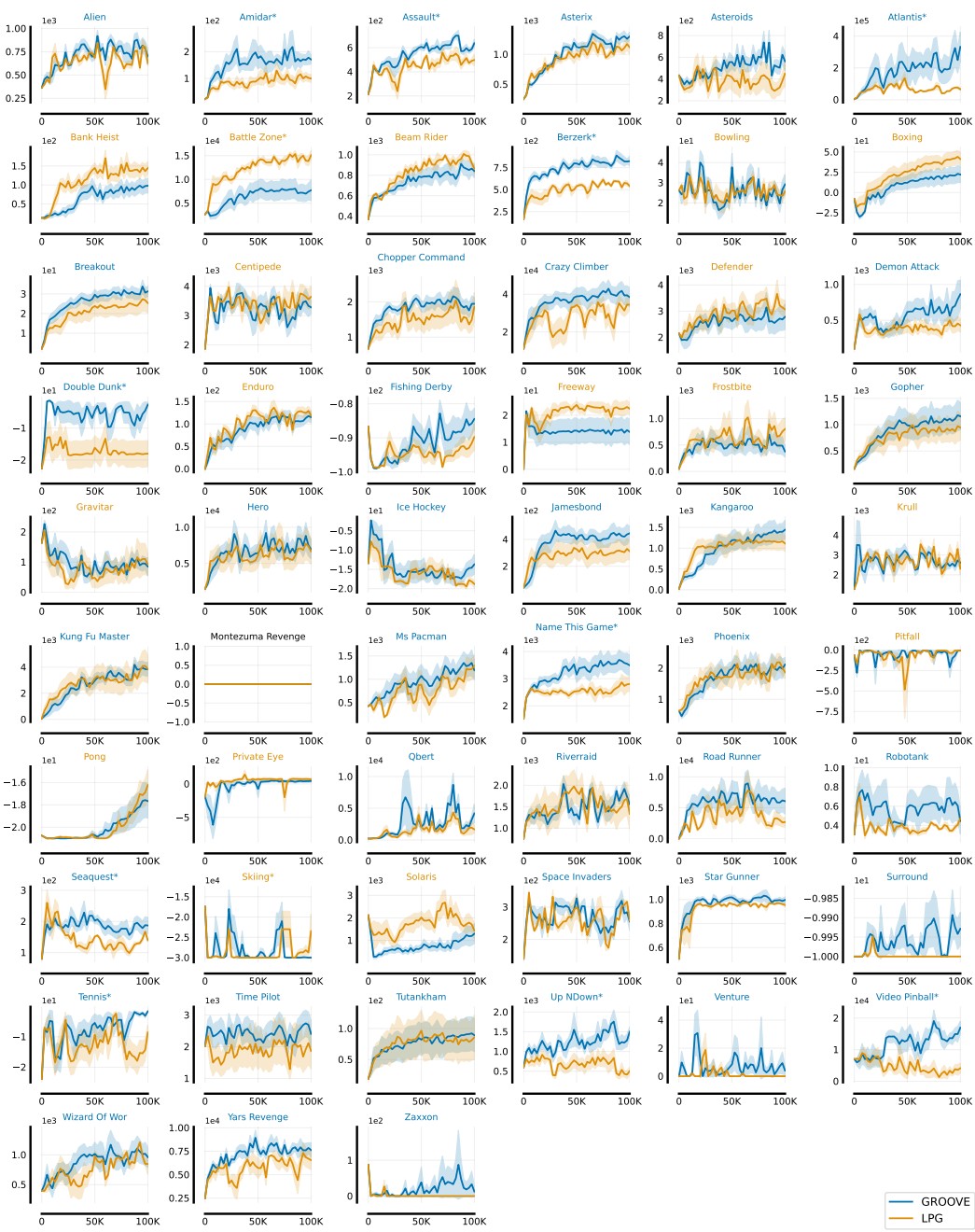

Figure 7: Atari training curves—environment names are highlighted according to highest evaluation return, asterisks (*) denote significant differences in evaluation return (5 seeds, $p < 0.05$).

# E    Min-Atar Per-Task Performance

As expected, we observe increased noise when breaking down performance by individual Min-Atar tasks, however, the results from the majority of tasks support our earlier conclusions. Firstly, we observe a significant positive correlation between the number of random training levels and return on three of the four Min-Atar tasks, again demonstrating the impact of task diversity on generalization. When controlling for the number of levels, we observe improved performance after training on handcrafted, rather than random, levels on three of the four Min-Atar tasks. Furthermore, on Asterix, training on handcrafted levels results in higher performance than the largest set of $2^{10} = 1024$ random levels, supporting our conclusion about level informativeness.

After training on high-AR levels, we observe an improvement against random levels on at least three of the four Min-Atar tasks for all sizes of training environment set up to $2^6 = 64$ levels. Beyond this, random and high-AR levels outperform each other on an equal number of tasks, however the dilution in mean AR for larger training sets makes this convergence unsurprising. Furthermore, high-AR levels are competitive with handcrafted levels at the same training set size and quickly outperform the fixed handcrafted set as more high-AR levels are added, demonstrating the effectiveness of AR at identifying informative curricula.

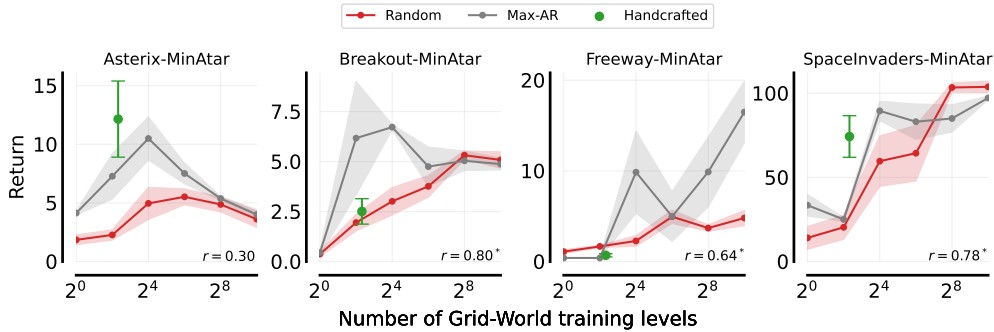

Figure 8: Generalization performance on Min-Atar, after meta-training LPG on variable-sized sets of Grid-World levels (5 seeds)—levels are selected through uniform-random sampling of all levels ("Random"), from the highest-regret levels of a previous LPG instance ("Max-AR"), or from a set of five handcrafted levels ("Handcrafted"). Pearson correlation coefficient is given for Random levels; significant positive correlations are marked with an asterisk (*).

# F    GROOVE vs. LPG Procgen Evaluation

After meta-training on Grid-World, we observe superior GROOVE performance on 2 out of 4 Procgen environments, superior LPG performance on 1 environment, and no difference on the remaining environment. We note that A2C is very weak on Procgen, failing to learn on the majority of environments, so we selected a subset of Procgen levels that A2C managed to learn in preliminary experiments. Procgen poses a robustness challenge that has required an extensive amount of further research to solve, using components not found in LPG or GROOVE.

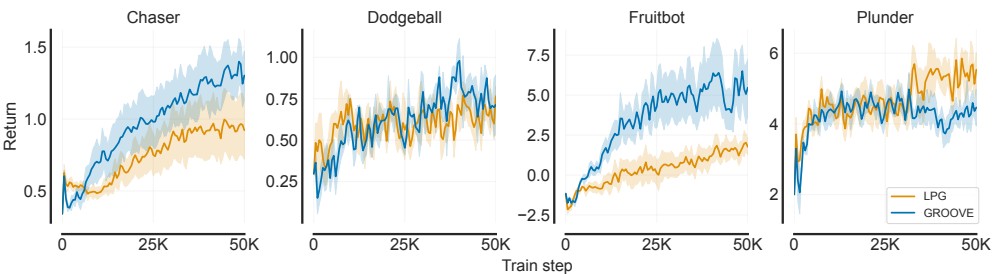

Figure 9: GROOVE and LPG training curves on Procgen (test performance, 5 seeds).

# G    Algorithmic Regret Antagonist Comparison

In order to investigate the impact of the antagonist agent on the performance of AR, we evaluated the performance of GROOVE with a range of antagonists (Figure 10). On Min-Atar, using a random or optimal agent as the antagonist for AR results in lower performance than using A2C or PPO on all environments. Furthermore, using A2C achieves higher performance than PPO on all environments.

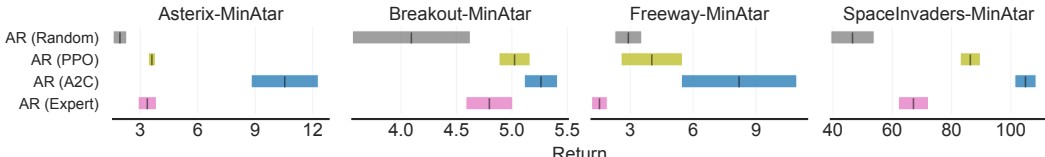

Figure 10: GROOVE Min-Atar performance after Grid-World meta-training, using random, expert, A2C and PPO agents as the algorithmic regret antagonist—mean return over 10 random seeds is marked, with standard error shaded.

To further investigate this result, we evaluated PPO and A2C on both random and difficult, handcrafted Grid-World levels. PPO achieves lower performance than A2C on Grid-World, with a larger gap on difficult, handcrafted Grid-World levels. This explains the previous results, as PPO will be inferior at identifying difficult levels when used as the AR antagonist. Furthermore, the update parameterized by LPG is capable of representing A2C, but not PPO. This implies that levels solvable by A2C should also be solvable by LPG, making them useful for training. In contrast, PPO may identify levels that cannot be solved without components found in PPO (clipping, mini-batch iterations) but not LPG.

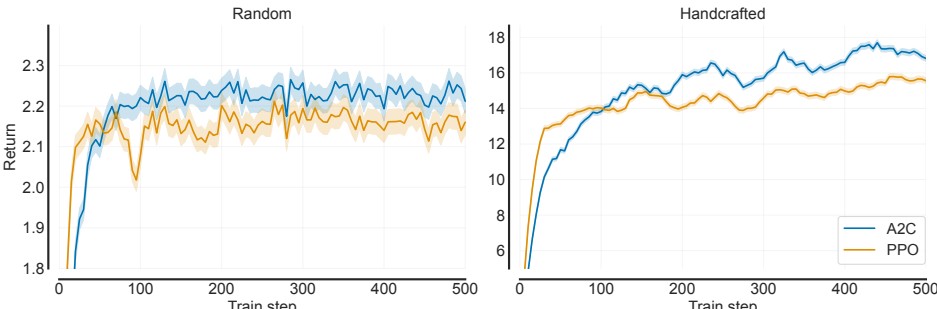

Figure 11: A2C and PPO training curves on random and handcrafted Grid-World levels—we observe a larger performance gap on harder, handcrafted levels (10 seeds).

