# OpenReview forum: "Discovering General Reinforcement Learning Algorithms with Adversarial Environment Design"
_NeurIPS.cc/2023/Conference — NeurIPS 2023 poster_

### Official Review · Reviewer_f1yc · 2023-07-06

**Soundness:** 3 good
**Presentation:** 3 good
**Contribution:** 2 fair
**Rating:** 6
**Confidence:** 3

**Summary:**

This paper focuses on the problem of learning update rules in reinforcement learning. By combining the advantages of the Learned Policy Gradient (LPG) algorithm and the Unsupervised Environment Design (UED) technique, the authors propose a meta RL algorithm, namely, GROOVE (General RL Optimizers Obtained Via Environment Design), to address the generalization problem for meta RL algorithms are applied to unseen environments. Accounting for the curricula generation of the meta-learned optimizer, the authors also design a metric named algorithmic regret (AR) to evaluate the regret during the meta-training, then be used to guide the learning of meta-learner. A series of experiments also present that GROOVE outperforms baselines on the Atari games.

**Strengths:**

1. This paper is well-organized, and the authors have a throughout introduction for the related work.
2. Experiments show that the designed AR outperforms existing metrics.

**Weaknesses:**

1. The approximation of algorithmic regret seems heavily relies on the choice of an oracle algorithm as UEDs, i.e., the choice of antagonist agent, which would be a performance bottleneck for GROOVE. Thus, there would be better if the authors demonstrate an ablation study on using different antagonist agents, i.e., training different RL algorithms.
2. The novelty is limited, as GROOVE is a direct combination of UED and LPG.

**Questions:**

Whether different antagonist agent will affect the performance?

---

> ### Author Rebuttal · Authors · 2023-08-09
>
> Thank you for your positive feedback about our writing and results. We have introduced **new results and edits in our summary response**, which we encourage you to read. Please find our response to each of your comments below:
>
> 1. Thank you for this suggestion, we agree that this is an interesting idea and have now performed this experiment, comparing random, expert, A2C, and PPO antagonists (**see new results**). The results verify the need for general RL algorithms as the AR antagonist and demonstrate how the choice of RL algorithm impacts generalization.
> 2. Whilst it is true that GROOVE builds on LPG and PLR, the primary message of our paper is that combining these existing methods is insufficient without our novel component (AR). This is demonstrated by our ablation study in Section 4.5, which shows the failure of existing metrics and the success of AR. To make this clearer, we have added a list of contributions, including this point, to the introduction (**see manuscript edits**).
> We also note that our PMO formalism, the application of environment design to PMO, and our analysis of how meta-training distributions impact generalization in this setting are entirely novel.
>
> Given the addition of your suggested experiment, along with our clarification and edit highlighting the paper’s novelty, we hope that you will consider raising your score to a stronger accept? Thank you!

---

> > ### Comment · Reviewer_f1yc · 2023-08-21
> > **Thanks for your response**
> >
> > I thank the authors for their response, and I'll keep my score as I do not think there is a significant novelty.

---

### Official Review · Reviewer_4F7T · 2023-07-07

**Soundness:** 2 fair
**Presentation:** 1 poor
**Contribution:** 3 good
**Rating:** 4
**Confidence:** 4

**Summary:**

The paper "Discovering General Reinforcement Learning Algorithms with Adversarial Environment Design" proposes an automatic curiculum approach for meta-learning of RL optimizers. It is based on a notion of regret of the optimizer, to choose environment parameters that are at a good level for the optimizer to leverage feedback from RL training steps. Since the notion of regret is intractable to be exactly computed, authors introduce a notion of algorithmic regret, which is an approximation. Results show intersting properties of the method.

**Strengths:**

- Meta-learning of RL optimizers is an important problem
- The proposal looks to contain innovative components
- Interesting results

**Weaknesses:**

- The paper is very hard, not to say impossible, to follow, as many many different components are not sufficiently defined (when not defined at all). To be useful, a research paper must contain every formalization and definition to allow an educated researcher to understand and reproduce the contribution. While I am not fully familiar with meta-learning of optimizers, I have a strong background in RL and I am still unable to put everything together in that paper to understand the proposal. I really think the paper should be fully rewritten before considering publication anywhere.  Here is a (non-exhaustive) list of things that are really difficult to understand:

            - U_\eta is introduced in 2.1 with y_t,\pi_t=U_\eta(x_t|x_t+1...x_T). I understand it is an LSTM on sequence x_t+1...x_T but why giving x_t| here ? Is it a distribution of x_t given the sequence ? This does not fit with the fact to feeding this to variables y and \pi. Is it the result of a LSTM step after feeding x_t in the sequence ? but x_t contains y_\theta(s_t) so it does not make sense to me.

             - In 2.1 we do not know how is trained U_\eta (\eta discussed only in 3.1) so it is very diffcult to understand what it captures.

            - what is y_\theta and why does it subscripts \theta as the policy \pi does ? policy and boostrap  share same parameters ? what is it supposed to capture ?

            - in 2.2 V gets a meta-optimizer F as input, in 3.1 it takes parameters and in 3.2 an RL algorithm

            - G in 3.1 is not defined : is it the regret ?

            - \cal{H} is not defined

            - F_\eta not well defined, we do not really know what controls \eta

            - Algo 1 : "Compute meta-gradient for \eta with updated \theta (eq 2)" ==> eq 2 is an expectation of expected return, not a gradient

            - Algo 1 : "update \Lambda with R and phi" => what does it concretely mean ? You store the regret associated with phi in the buffer ?

            - Algo 1: What means initialize lifetimes? How is it done ?




**Questions:**

see weakness above

**Limitations:**

..

---

> ### Author Rebuttal · Authors · 2023-08-09
>
> Thank you for your kind words about our problem, method and results. We appreciate the time you have taken to give such extensive feedback regarding our formalization. We address each of your concerns below and refer you to the **edits described in our summary response**, which we believe will resolve much of the stated confusion. If you have time, we encourage you to revisit the manuscript’s formalism after reading our edits and clarifications, which we hope will lead you to the same clarity and ease of understanding described by the other reviewers.
>
> * Thank you for spotting this, there is a mistake where the bootstrap vectors are not subscripted with \theta, which is likely the source of the confusion (**see line 77 edit**). However, there’s a minor mistake in your question, since the output of LPG is “\hat{y}\_t, \hat{\pi}\_t”, not “y\_t, \pi\_t”. Hopefully, this answers your questions since, as we state, “LPG outputs targets \hat{y}\_t, \hat{\pi}\_t” for y\_t and \pi\_t (targets being denoted by the hat), which are distinct from the agent outputs contained in x\_t, being the “probability of the chosen action \pi\_theta(a\_t|s\_t) and bootstrap vectors for the current and next states y\_\theta(s\_t), y\_\theta(s\_{t+1})”.
> * Regarding the | notation in U_\eta(x\_t|x\_{t+1}...x\_T), you are correct that this is done to reflect the reverse-recurrent processing of tokens in LPG. Without this constraint, this function could be generalized to U\_\eta(x\_t,x\_{t+1},...x\_T). However, we believe this notation is instructive for understanding the working of LPG, which processes each token individually based on the recurrent state from all future tokens.
> * We have now included the definition of \eta, in order to make clear that it parameterizes LPG (**see line 74 edit**). On lines 68-71, we present an intuitive interpretation of the information learned by \eta, followed by formally defining the function it parameterizes on lines 74-78. The loss for \eta is presented immediately after in the following subsection (Equation 2) once we have formally defined the PMO problem setting in the context of UED. We hope this makes \eta clearer; if not, is there any other information you believe would do so?
> * This confusion also likely results from our omitted definition of \eta (**see line 74 edit**). y\_\theta is defined as the function producing “bootstrap vectors for the current and next states y\_\theta(s\_t) and y\_\theta(s\_{t+1})”, which share the “agent parameters \theta” with \pi\_theta. On line 68-71, we provide an explanation of bootstrap vectors and their relation to critics in RL, making clear the information they capture.
> * Whilst this is a common abuse of notation, we already make a point of explaining the overloading of function V each time it is performed (lines 135-136 and 149) to avoid any confusion.
> * This is a good spot, thank you. We have added a definition of G to our edits (**see line 135 edit**).
> * \cal{H} is a capital \eta, denoting the space of the variable \eta. Whilst this use of capitalization for parameter space is common, we acknowledge that capital \eta is uncommon so we have added an explicit definition on line 135 in our edits to avoid confusion (**see line 135 edit**).
> * \cal{F} is defined as “a meta-optimizer \cal{F}: \Theta \times \bb{T} -> \Theta” on line 95. The statement “the update rule parameterized by \eta” on lines 136-137 and our new definition of \eta (**see line 74 edit**) should now make its meaning clear.
> * Algorithm 1 is presented in pseudocode and intended to give a high-level overview of GROOVE’s meta-training, much like Figure 2. This is done to improve the reader’s ease of understanding the method since exhaustively defining all operations here would make the algorithm opaque. All algorithm components are individually detailed in greater depth elsewhere in the manuscript. With this in mind, we respond to each of your related comments below:
>     * It is true that Equation 2 is not a gradient. However, this reference is intended to contextualize a line of non-formal pseudocode rather than formally define the gradient computation, and it trivially follows that the meta-gradient is the gradient of the meta-objective given in Equation 2, so we do not believe that restating it separately adds clarity.
>     * That’s right, the update primarily stores the regret associated with phi in the buffer. However, the entire PLR procedure is more involved than that and would make the algorithm significantly more verbose to include in its entirety (see the original PLR work, [1]). Therefore we define \Lambda as the “PLR level buffer” in the algorithm rather than outlining the entire PLR procedure here.
>     * Initializing a lifetime involves sampling the level \phi from the “PLR level buffer \Lambda” and the agent parameters \theta from their “initialization function p(\theta)”. This is presented in the algorithm, with the line “Reinitialise lifetime (\phi, \theta) ~ \Lambda \times p(\theta)”.
>
> We thank you again for taking the time to find these oversights and hope that our edits and clarifications resolve any remaining confusion. In light of these and the opinion of the other reviewers that the paper is “well-structured/organized” (qWd8, f1yc), “easy to understand” (qWd8), and “provides a clear explanation of the underlying concepts” (TbjH), we wonder if you would be willing to increase your score in order to reflect the contributions that you mention?
>
> [1] M. Jiang, E. Grefenstette, and T. Rocktäschel. Prioritized level replay. In International Conference on Machine Learning, pages 4940–4950. PMLR, 2021

---

> > ### Comment · Reviewer_4F7T · 2023-08-11
> > **Thanks**
> >
> > Thanks to authors for their insightful answers that help le better understand the contribution.
> >
> > Despite this and ither revierwers opinions,  still think the paper should be reorganized to help the reader. At least the global objective should be clearly formalized at the begining of section 2. Maybe 2.2 before 2.1 would be easier to read ? There is also still no clear definition of what is a lifetime that is core of the algorithm. I understand arguments if authors that claim that algo1 is high level picture, but I feel that in that form it does not help. Giving some further details in it would be very insightful. Also, a picture that would include all notation to illustrate the learning process would be very helpful (figure 1 is not).
> >
> > I still think that this paper, even if I am sure present very interesting contribution that I would like to see published soon, is not ready for publication at this round. Although, I won't oppose to its acceptance if AC and other reviewers like it. Thus, I increase my score to bordeline reject to reflect this.

---

> > > ### Author Response · Authors · 2023-08-12
> > > **Response and further additions**
> > >
> > > Thank you very much for your prompt reply and for taking our rebuttal into consideration!
> > >
> > > We appreciate and agree with your remaining feedback about the paper’s structure. Therefore, **we have implemented all of your remaining suggestions** for the camera-ready copy (or next revision). Namely,
> > > * *Section 2*: Move the problem formulation (lines 61-66 and 80-105) to 2.1 and the LPG background (lines 67-78) to 2.2,
> > > * *Line 98*: Extend the definition of a lifetime to include the task \phi and the agent parameters \theta,
> > > * *Algorithm 1*: Reference the relevant paper section for each operation (to maintain the algorithm’s simplicity whilst making further details easy to locate),
> > > * *Supplementary materials*: Add a summary of the paper’s notation.
> > >
> > > We hope these edits sufficiently address the remainder of your feedback. Following their implementation in the camera-ready copy, we are confident that the paper is ready for publication. If you disagree, **are there any outstanding concerns keeping the paper below the acceptance threshold?**
> > >
> > > Thank you again for your feedback, it has greatly improved the paper’s clarity!

---

### Official Review · Reviewer_TbjH · 2023-07-14

**Soundness:** 3 good
**Presentation:** 3 good
**Contribution:** 2 fair
**Rating:** 7
**Confidence:** 2

**Summary:**

The paper presents an approach, i.e. GROOVE, to learning an update rule for generalization on unseen tasks automatically,  based on the idea of Unsupervised Environment Design, where a student agent is trained on an adaptive distribution of environments proposed by a teacher and the teacher seeks to propose tasks which maximize the student’s regret. The authors also introduce a new concept of algorithmic regret, which is used to approximate regret and automatically generate curricula. The results of experiments comparing GROOVE and LPG demonstrate the superiority of GROOVE in terms of generalization.

**Strengths:**

1. The proposed method is well-motivated and the authors provide a clear explanation of the underlying concepts.
2. GROOVE, especially the use of algorithmic regret, is simple to understand and effective that is well-supported by the experimental results and analysis.

**Weaknesses:**

1. The paper could benefit from listing its contributions, e.g. in the first chapter, which helps readers capture its novelty faster.
2. More detailed discussion of the connection between GROOVE and related works in the field of RL and meta-learning (i.e. in the part "Alternative Approaches to Meta-Reinforcement Learning")  will be helpful for readers to address this work.

**Questions:**

1. Have the authors tried other RL algorithms as the baseline in the Algorithmic Regret? if so, how do they perform?

**Limitations:**

as stated in the weakness

---

> ### Author Rebuttal · Authors · 2023-08-09
>
> Thank you for your feedback and overall positive review of our work. We have provided **edits and new results in our summary response**, which we encourage you to read. Please find our response to the weaknesses below:
> 1. We thank you for this suggestion and agree that listing contributions would be an effective format, so have **added a contribution list in our edits**.
> 2. Since GROOVE is based on ideas from PMO and UED, we discuss its connection to each of these fields in the “Policy Meta-Optimization” and “Unsupervised Environment Design” sections of the Related Work. Here, we highlight GROOVE’s improved robustness and generalization performance on OOD tasks. Regarding the wider field of meta-RL, we contrast this with the PMO problem setting as a whole, rather than just GROOVE, in the “Alternative Approaches to Meta-Reinforcement Learning” section. This avoids us directly comparing GROOVE to methods outside of PMO, which aim to solve different problems.
>
> In response your question, suggesting that we evaluate alternative RL algorithms for Algorithmic Regret (AR), we have now performed and included this experiment (**see new results**). We thank you for this suggestion, as we believe it demonstrates interesting properties of AR: that providing the optimal policy or a random agent is insufficient and can select artificially difficult or easy levels, whilst the choice of RL algorithm and its capability on the meta-training distribution also influences generalization.
>
> We hope that the addition of your suggested experiment has provided the insights you hoped for, and that our edit and clarifications have sufficiently addressed both of your concerns. If so, would you consider increasing your rating and/or contribution scores? Thank you!

---

### Official Review · Reviewer_qWd8 · 2023-07-19

**Soundness:** 3 good
**Presentation:** 3 good
**Contribution:** 2 fair
**Rating:** 5
**Confidence:** 2

**Summary:**

The authors present a meta-learning method to enhance an RL algorithm's performance across diverse RL tasks. They utilize the concept of unsupervised environment design (UED) methods for training RL agents and adapt this approach for the task of meta-learning a policy optimizer. The authors propose a novel algorithmic regret (AR) to facilitate UED and confirm its superior performance to L1 value loss and positive value loss through experiments. The experiments verify the effectiveness of proposed method.

**Strengths:**

1. The paper is well-structured and easy to understand.
2. A thorough review of related studies is provided.
3. The proposed idea, while simple, appears logically sound. However, as I am not intimately familiar with the domain of reinforcement learning, I am unable to evaluate its novelty.

**Weaknesses:**

1. Could the authors further elaborate why the proposed Algorithmic regret is better than the approximation method before? Some intuitive discussions are necessary in addition to only experimental results. Moreover, AR is compared to L1 value loss and positive value loss in Sec. 4.5. It seems there are also some other loss  in Lines 16-25. Could the authors clarify the reason that the comparison between AR and these loss is not conducted?

2. It would be informative if the authors could discuss the scenarios depicted in Fig. 1 where Groove performs less effectively than LPG.

3. "The result is our method" in abstract could be rephrased.

4. It is mentioned that "By examining how characteristics of the meta-training distribution impact the generalization of these algorithms". It is unclear to me where it is reflected in the paper. Could the authors further illustrate it?

Minor:
Line 50 Section 4.4 -> Section 4.5

**Questions:**

Please refer to the 'Weaknesses' section above for questions and clarifications.


**Limitations:**

No.

---

> ### Author Rebuttal · Authors · 2023-08-09
>
> Thank you for your supportive review of our method, evaluation, and clarity. In response to your comments, we have provided **new results in the author rebuttal**. Please find our response to each of your comments below:
> 1. We have now added a comparison of A2C, PPO, expert, and random antagonist agents in AR, in order to further explain the role of the antagonist on AR’s performance (**see new results**). Additionally, we hope the simplicity of the method and the interpretation of AR as a proxy for informative levels (see Section 4.3) allows the reader to form a strong intuition for how it achieves its performance. Beyond this, we do not provide speculative interpretations for why AR succeeds in the paper, but we hypothesize that levels identified by AR will reflect the biases of the antagonist agent. Therefore, using generalizable algorithms (A2C, PPO) rather than specialized (expert) or weak (random) algorithms as the antagonist should identify levels with common properties.
> Regarding the omitted regret metric (maximum Monte Carlo) mentioned on line 124, we do not evaluate against this baseline since it is only a minor adaptation of positive value loss, designed to handle sparse-reward environments [1]. Since we meta-train on Grid-Worlds, where objects frequently respawn and can be reached in a small number of steps, we do not require this adaptation and therefore use the original form of positive value loss.
> 2. Understanding the properties of optimizers learned by GROOVE and LPG is an interesting and important line of work. Since the application of environment design to PMO is an entirely novel topic, the focus of this work is to extensively demonstrate its potential to improve generalization performance. To achieve this, we compare both in-distribution robustness (Figure 5) and out-of-distribution performance on Min-Atar (Figure 6), Atari (Figures 1 and 4) and now Procgen (**see new results**), providing a comprehensive demonstration of GROOVE’s effectiveness. Since these benchmarks are not designed to test interpretable capabilities of agents, we were unable to find any common theme between the settings on which GROOVE outperformed LPG. However, we plan on performing this analysis in future work.
> 3. Thank you for pointing this out, we see how it could be unclear. We will rephrase this sentence.
> 4. We perform this analysis in Section 4.2, “Designing Meta-Training Distributions for Generalization”. In this, we evaluate the OOD performance of LPG whilst manually controlling the diversity and informativeness of meta-training tasks, thereby relating characteristics of the meta-training distribution to the algorithm’s generalization performance.
>
> We hope that we have clarified each of your questions and will be sure to edit the sentence discussed in Q3. Please let us know if any of these points remain unclear, or if you have further questions. If not, in light of these responses, our new results, and your praise of our method, evaluation, and clarity, would you consider upgrading your score above a borderline? Thank you!
>
> [1] M. Jiang, M. Dennis, J. Parker-Holder, J. Foerster, E. Grefenstette, and T. Rocktäschel. Replay guided adversarial environment design. Advances in Neural Information Processing Systems, 34: 1884–1897, 2021

---

> > ### Comment · Reviewer_qWd8 · 2023-08-21
> > **Thank you**
> >
> > Thank you for your response to my comments. I acknowledge the efforts made to address the questions I raised. Because I am not quite familiar with the reinforcement learning topic, I will maintain my score and be attentive to the other reviewers' thoughts.

---

### Official Review · Reviewer_8DkV · 2023-08-02

**Soundness:** 2 fair
**Presentation:** 2 fair
**Contribution:** 3 good
**Rating:** 5
**Confidence:** 5

**Summary:**

This paper proposed a new framework, GROOVE, to solve a new Meta-UPOMDP problem. It also introduces algorithmic regret (AR) to approximate the regret to update the curator and generator (sampler?).  The results show that the meta-optimizer learned by GROOVE can be used to improve the games in Atari.

**Strengths:**

1. The proposed problem Meta-UPOMDP is important.
2. The experiments show that the method can work even from grid-world to atari.
3. The AR idea is novel.
4. The results of AR in Fig. 6 is great.

**Weaknesses:**

1. Several definitions are absent, including the sampler in Fig2 and the symbol \eta mentioned in line 74.
2. It would be beneficial to introduce PMO earlier. The headline and abstract might lead readers to expect traditional RL and few-shot meta-learning.
3. It is recommended to conduct tests on additional environments, such as Procgen. Even just for evaluating the meta-optimizer learned in Grid-World.
4. Since AR constitutes a crucial aspect of the work, further insights should be provided.
5. The majority of the work involves combining existing works (LPG and PLR).

**Questions:**

1. How do you select algorithm A in equation 4? Including the hyperparameter of the algorithm.
2. The claim in 209 seems similar to [1]. Please explain the difference.
3. What is the probability of using the generator and the curator?



[1] Li, Y., & Zhang, C. (2022). On Efficient Online Imitation Learning via Classification. Advances in Neural Information Processing Systems, 35, 32383-32397.

**Limitations:**

Mentioned in weakness.

---

> ### Author Rebuttal · Authors · 2023-08-09
>
> We thank you for your kind words regarding the problem’s importance, the novelty of algorithmic regret (AR), and the demonstrations of its performance. We encourage the reviewer to **read the edits and new results outlined in the author rebuttal**. We respond to each of your comments below:
>
> Weaknesses:
> 1. Thank you for pointing these out, we have now added the definition of \eta (**see line 74 edit**). The use of “sampler” rather than “generator” (see Section 3.3) reflects that this component generating random levels in PLR, rather than learning a generative model. We will highlight this distinction in the caption.
> 2. We agree that the distinction between PMO and traditional meta-RL is important. No formal distinction between the two settings has been proposed in prior work, which is why we introduce PMO as a novel problem formulation in this work. Since readers will be unfamilar with this terminology, we use existing terminology - such as “discovering RL algorithms” in the title and “meta-learn update rules” in the abstract - to describe the setting before mentioning it in the introduction and formalizing it in Section 2.1.
> 3. This is a good suggestion and we have now added your suggested experiment of GROOVE vs. LPG on Procgen (**see new results**). We chose to prioritize Atari for our original evaluation to be consistent with the environments used in the original LPG paper, but we hope this provides the additional insights you hoped for.
> 4. We have now added a further experiment for AR, evaluating the impact of different antagonist agents on generalization performance (**see new results**). In addition to this, we believe Figure 3 provides strong insight into AR as a proxy objective for the informativeness of levels, whilst Figure 6 directly compares it to existing metrics, demonstrating its effectiveness.
> 5. Whilst it is true that GROOVE builds on LPG and PLR, the primary message of our paper is that these existing methods fail to work together without our novel component, AR. This is demonstrated by our ablation study in Section 4.5, which shows the failure of existing metrics and success of AR. To make this clearer, we have added a list of contributions, including this point, to the introduction (**see manuscript edits**).
>
> Questions:
> 1. For generality, we define AR to be agnostic to the underlying RL algorithm. In our experiments, we selected A2C due to it being used by LPG at meta-train time, using existing hyperparameters for A2C from prior work. We detail these in Table 2 of the supplementary material, but will add a clarification of the source of our hyperparameters there.
> 2. Thank you for referring us to this work, the similarity is very subtle. The foremost difference is problem setting: they study algorithms for online imitation learning, rather than policy meta-optimization algorithms. In addition to this, they analyze sample complexity in the number of interaction rounds, expert annotations and oracle calls, whilst we fix the number of environment interactions (samples) and control the number of training environments.
> 3. This can be found in Table 1 of the supplementary material. The probability of using the curator (known in UED literature as “replay probability”) is a tunable hyperparameter of the algorithm. We use 0.5, which we selected from preliminary tuning and prior PLR implementations.
>
> In summary, we hope that the referenced edits have resolved Weaknesses 1 and 5, our new results resolve Weaknesses 3 and 4, and that we have provided sufficient clarification for Weakness 2 and your questions. We thank you again for your praise of the importance, novelty, and evaluation of this work. In light of our response, would you consider increasing your score to an accept? Thank you!

---

> > ### Comment · Reviewer_8DkV · 2023-08-21
> > **Official comment**
> >
> > 1. Thank you for providing the results of Procgen. However, the experiments do not train for enough steps. Normally, Procgen needs to train for 200m steps. Hence, it is hard to tell about the improvement.
> > 2. Thank you for providing experiments on using different algorithms as regret antagonists. However, it seems like most algorithms are bad besides A2C. These results lowered my confidence in the method.

---

> > > ### Author Response · Authors · 2023-08-21
> > > **Clarifications**
> > >
> > > 1. Our results show Procgen training for **200M environment steps**, which is equivalent to **50K train steps** (agent updates). Therefore, we do follow the standard procedure.
> > > 2. These results are entirely consistent with our hypothesis: AR with A2C or PPO as the baseline **outperforms specialized algorithms on every environment and all other alternative regret measures on 3/4 environments**. Furthermore, due to the time allowed for rebuttal, we were unable to fully tune our PPO implementation. Despite this, we hypothesize that PPO would still underperform A2C due to it having a different functional form than LPG, which motivated the original choice of A2C.

---

> > > > ### Comment · Reviewer_8DkV · 2023-08-21
> > > > **Official comment**
> > > >
> > > > If you have trained for 200M, then the return is way lower than normal PPO. Is it because you are using A2C?

---

> > > > > ### Author Response · Authors · 2023-08-21
> > > > >
> > > > > That's correct, we're using A2C. Procgen poses a difficult generalization problem that has required significant research beyond A2C (including PPO) to solve, making this performance unsurprising. However, for consistency with our other experiments and LPG, we compare to A2C.

---

> > > > > > ### Comment · Reviewer_8DkV · 2023-08-22
> > > > > > **Official comment**
> > > > > >
> > > > > > I don't have any additional questions. I'm inclined to accept this paper, contingent on the consensus of the other reviewers.

---

### Author Rebuttal · Authors · 2023-08-09

Thank you to all of the reviewers for their detailed and insightful feedback.

We appreciate the positive comments describing our problem setting as **important** (8Dkv, 4F7T) and **well-motivated** (TbjH), in addition to our proposed method containing **innovative components** (4F7T), with Algorithmic Regret (AR) being described as **novel/new** (8DkV, qWd8, TbjH). Furthermore, we are thankful that **all reviewers provide positive feedback regarding our results**, describing them as great (8DkV) and interesting (4F7T), whilst showing the method is well-supported (TbjH), achieves superior performance to existing metrics (qWd8, f1yc) and can even [transfer] from Grid-World to Atari (8DkV). Finally, we thank the reviewers for their kind words regarding the paper’s writing, specifically that it is **well-structured/organized** (qWd8, f1yc) and **easy to understand** (qWd8), providing a **clear explanation** of the underlying concepts (TbjH) and a **thorough introduction** to related work (qWd8, f1yc).

## Manuscript edits
* Reviewers 8DkV and 4F7T pointed out omissions in our formalism. As a result, we have **added the following correction and new definitions** to rectify their concerns,
    * *Line 74*: Define \eta as the meta-parameters of LPG,
    * *Line 77*: “y(s_t) + y(s_{t+1})” => “y_\theta(s_t) + y_\theta(s_{t+1})”,
    * *Line 135*: Define G as expected return and capitalized \eta as the space of the meta-parameters, \eta.
* Reviewers 8DkV and f1yc raised a concern over our proposed method, GROOVE, containing existing components from UED and PMO. Firstly, the application of UED to PMO is already a novelty. Furthermore, our results underscore that combining existing methods (PLR and LPG) is insufficient without our novel component (AR). In addition to this, reviewer TbjH suggested that we list contributions in the introduction to help readers capture this and other novelties faster. Therefore, we add the following **list of contributions** to the end of the introduction (*line 58*):
    * In order to distinguish this problem setting from traditional meta-RL, we provide a novel formulation of PMO using the Meta-UPOMDP (Section 2).
    * We propose AR (Section 3.2), a novel regret approximation for PMO, and GROOVE (Section 3.3), a PMO method using AR for environment design.
    * We analyze how features of the meta-training distribution impact generalization in PMO (Section 4.2) and demonstrate AR as a proxy for task informativeness (Section 4.3).
    * We comprehensively evaluate GROOVE against LPG, demonstrating improved in-distribution robustness and out-of-distribution generalization on Min-Atar, Atari, and Procgen (Section 4.4).
    * We perform an ablation of AR, our novel component, demonstrating the insufficiency of existing methods (PLR and LPG) without AR, as well as the impact of the antagonist agent in AR (Section 4.5).
    * We release our implementation of GROOVE, PLR and LPG, capable of single-GPU meta-training in 3 hours.

## New results (see PDF)
* Reviewers TbjH and f1yc suggested an **ablation of the antagonist agent in AR** to determine its impact on the method’s effectiveness, whilst reviewer 8DkV asked for further insights about AR. In response to both of these, we have performed the suggested ablation and will add it to Section 4.5.
    * *Figure 9*: On Min-Atar, using a random or optimal agent as the antagonist for AR results in lower performance than using A2C or PPO on all environments. Furthermore, using A2C achieves higher performance than PPO on all environments.
    * *Figure 10*: PPO achieves lower performance than A2C on Grid-World, with a larger gap on difficult, handcrafted Grid-World levels. This explains the previous results, as PPO will be inferior at identifying difficult levels when used as the AR antagonist. Furthermore, the update parameterized by LPG is capable of representing A2C, but not PPO. This implies that levels solvable by A2C should also be solvable by LPG, making them useful for training. In contrast, PPO may identify levels that cannot be solved without components found in PPO (clipping, mini-batch iterations) but not LPG.
* Reviewer 8DkV suggested we further evaluate **GROOVE vs LPG on Procgen**, which we have now included and will add to Section 4.4.
    * *Figure 11*: After meta-training on Grid-World, we observe superior GROOVE performance on 2 out of 4 Procgen environments, superior LPG performance on 1 environment, and no difference on the remaining environment. We note that A2C is very weak on Procgen, failing to learn on the majority of environments, so we selected a subset of Procgen levels that A2C managed to learn in preliminary experiments. Procgen poses a robustness challenge that has required an extensive amount of further research to solve, using components not found in LPG or GROOVE.

In summary, we have introduced edits to address the reviewers’ concerns regarding writing and novelty, in addition to new results to provide further insights about our method. In light of these, we welcome any further feedback and hope the reviewers will consider raising their original scores.

---

### Author Response · Authors · 2023-08-17
**Disappointing engagement**

Unfortunately, after a week of the discussion period, we received only a single response to our rebuttal (thank you to reviewer 4F7T).

**Please may reviewers 8DkV, qWd8, TbjH, and f1yc take a look at our rebuttal**, which includes multiple new results and edits, and consider updating their scores if we have adequately addressed their feedback?

---

### Decision · Program_Chairs · 2023-09-21

**Decision:**

Accept (poster)

**Comment:**

This paper is concerned with enhancing the performance of reinforcement learning algorithms across diverse tasks using a meta-learning approach. The proposed method, called GROOVE, incorporates unsupervised environment design and introduces algorithmic regret to facilitate task generalization, resulting in superior performance compared to LPG in terms of generalization ability. The idea looks interesting and novel, and the proposed method is sound with good experimental support.